# Non-menstrual pelvic symptoms and women's quality of life: a cross-sectional observational study

Marlon de Freitas Fonseca[1]*, Felipe Ventura Sessa[2], Claudio Peixoto Crispi Jr[3], Nilton de Nadai Filho[1‡], Bruna Rafaela Santos de Oliveira[4‡], Rafael Ferreira Garcia[2‡], Claudio Peixoto Crispi[3#]

**1** Instituto Nacional de Saúde da Mulher, da Criança e do Adolescente Fernandes Figueira (Fiocruz), Rio de Janeiro, RJ, Brazil, **2** Instituto de Psiquiatria da Universidade Federal do Rio de Janeiro, Rio de Janeiro, RJ, Brazil, **3** Instituto Crispi de Cirurgia Minimamente Invasivas, Rio de Janeiro, RJ, Brazil, **4** Hospital Universitário Clementino Fraga Filho, Universidade federal do Rio de Janeiro, Rio de Janeiro, RJ, Brazil

☙ These authors contributed equally to this work.
‡ These authors also contributed equally to this work.
#*In memoriam*.
* marlon.iff@gmail.com

## Abstract

### Background

Endometriosis-related pain duration and burden are associated with poorer psychological well-being and health-related quality of life (HQoLife). This study aimed to compare the relative burdens of 4 different non-menstrual pelvic symptoms on HQoLife: Deep dyspareunia (DDyspareunia), non-menstrual pelvic pain (NMPelvicPain), non-menstrual dyschezia (NMDyschezia) and non-menstrual dysuria (NMDysuria).

### Subjects and Methods

This is a pre-planned cross-sectional interdisciplinary retrospective observational study. The sample consists of 369 consecutive patients referred for minimally invasive surgery at a private institution. DDyspareunia, NMPelvicPain, NMDyschezia and NMDysuria were assessed with a self-reported 11-point (0–10) numeric rating scale (NRS). The Short Form 36 (SF36) and Endometriosis Health Profile 30 (EHP30) full questionnaires were applied to assess HQoLife. Multiple linear regression models were used to compare the 4 explanatory variables (correlates), which were tested for 19 different HQoLife domains.

### Results

Multivariate exploratory analyses indicated that NMPelvicPain and NMDyschezia may be the non-menstrual pelvic symptoms that most impact the HQoLife of women with endometriosis, an association that was observed in practically all domains of

**Data availability statement:** All relevant data are within the paper and its Supporting Information files.

**Funding:** Publication fee was funded by a Brazilian Government Institution (CAPES-PROAP budget – award number 001/2021 VPEIC / Fiocruz - Programa de Pós-graduação em Pesquisa Aplicada à Saúde da Criança e da Mulher do Instituto Nacional de Saúde da Mulher, da Criança e do Adolescente Fernandes Figueira, da Fundação Oswaldo Cruz). Fiocruz provided support in the form of salaries for the author MFF, but did not have any additional role in the study design, data collection and analysis, decision to publish, or preparation of the manuscript. We also declare that our study received no funding from any company or institution for the field work and development, all costs for this were due to the authors.

**Competing interests:** None of the authors have any competing interest that could be perceived to bias this work. In the same way, there was no influence of any of the financial support in the results presented.

both the SF36 and the EHP30 questionnaires. DDyspareunia was the most important symptom for the sexual intercourse EHP30 domain. Despite the biological plausibility and statistical significance found in virtually all models, their low explanatory power suggests the existence of additional major covariates (not contemplated in this study). Moreover, the presence of significant positive intercepts (P<0.001) implies that some HQoLife impairment is expected for any hypothetical woman with endometriosis, even if all 4 correlates' scores are equal to zero. Our findings support the hypothesis that the importance of bowel symptoms has been underestimated in endometriosis assessment whereas the burden of other potential covariates should not be neglected.

## Conclusion

DDyspareunia was the most important non-menstrual pelvic symptoms for the EHP-30Sex domain, but NMPelvic Pain and NMDyschezia were the most important for overall HQoLife.

## Introduction

Endometriosis is an inflammatory condition of multifactorial etiology. It is highly prevalent, affecting roughly 10% of reproductive-age girls and women globally (190 million) [1]. The most severe form – Deep Infiltrating Endometriosis or just Deep Endometriosis – refers to endometriosis that infiltrates one or multiple organs to depths exceeding 5 mm [2,3]. Endometriosis frequently causes one or more pelvic dysfunctions, including pain and infertility, which impair women's life [4,5,6].

Potentially influenced by biological, psychological and social factors, pain may be defined as an unpleasant sensory and emotional experience associated with, or resembling that associated with, actual or potential tissue damage [7]. Quality of life may be defined as individuals' perceptions of their position in life in the context of the culture and value systems in which they live and in relation to their goals, expectations, standards and concerns; this definition reflects the view that quality of life refers to a subjective evaluation that is embedded in a cultural, social and environmental context [8]. Health-related quality of life (HQoLife) is generally considered a multidimensional assessment of how disease and treatment affect a patient's sense of overall function and well-being [9].

From an evolutionary perspective, the increased lifetime exposure to menstruation due to earlier menarche, lower fecundity, and later mortality may, through hyperalgesic priming, provide the basis for severe menstrual cyclic symptoms to extend beyond the menstrual period or even to shift into a chronic pelvic pain state [10]. Distinct from the typical endometriosis-related cyclic symptoms (dysmenorrhea, cyclic low back pain, cyclic dyschezia, cyclic bloating, etc.) [11], non-menstrual pelvic pain is a common and complex acyclic condition whose cause is often clinically inexplicable; it is influenced by emotional aspects and usually impacts woman's life unpredictably

[12]. Actually, the acyclic symptoms may occur at any time of the month or on a daily basis; they include deep dyspareunia (DDyspareunia), non-menstrual pelvic pain (NM Pelvic Pain), non-menstrual dyschezia (NM Dyschezia) and non-menstrual dysuria (NM Dysuria) [13]

Endometriosis is reported to have an adverse impact on physical, mental, and social well-being, with a negative effect on HQoLife [14,15,16]; patients may present significant psychopathological comorbidities – especially high levels of somatization, depression, sensitivity, and anxiety [17]. Moderate/ severe hallmark endometriosis-related symptoms were associated with worse HQoLife, with the greatest impact from NM Pelvic Pain [18]. Women suffering chronically from NM Pelvic Pain experience higher rates of emotional dysregulation (anxiety or depression) than those with cyclic pain [19]. While cyclic symptoms such as dysmenorrhea mainly impair physical quality of life, NM Pelvic Pain is the symptom most associated with increased anxiety and depression and with worse quality of life and mental health [20].

Endometriosis-related pain duration and burden are associated with poorer psychological well-being and HQoLife, and women's life can be improved after eliminating even one endometriosis symptom [21,22]. Surgery remains the treatment of choice to improve HQoLife in cases in which medical management has been ineffective for pain relief [23,24,25] or in selected cases of endometriosis-related infertility [26]. Even for these indications, the benefits of minimally invasive endometriosis surgery must be weighed against the known risks of the urinary [27] and bowel [28] complications, and a shared and informed decision is essential to achieve the best outcome [29].

It has increasingly been believed that endometriosis must be treated with a multidisciplinary vision that includes not only a medical approach but also psychological, work, and economic support [30]. Exploring the complexity and variability in how women with endometriosis experience their pain may help researchers to understand which interventions might be helpful to patients and provide clinicians with more understanding about how different endometriosis-related symptoms are for different women [13]. Focusing on non-menstrual pelvic symptoms, this study used exploratory multivariate analysis to compare the relative burdens of DDyspareunia, NM Pelvic Pain, NM Dysuria and NM Dyschezia on HQoLife.

## Materials and methods

### Design and setting

This is a pre-planned cross-sectional interdisciplinary retrospective observational study exploiting a long-established database of clinical information collected about each of our endometriosis patients. The sample consists of 369 consecutive patients referred by the patient's personal gynecologist to the Crispi Institute for Minimally Invasive Surgery (CRISPI) – a private institution located in Rio de Janeiro, RJ, Brazil – from January, 1 2018 through July, 30 2021 for consideration of minimally invasive surgical treatment of deep endometriosis for infertility and/or pain persisting after medical management.

The Strengthening the Reporting of Observational Studies in Epidemiology (STROBE) statement guidelines [31] were followed to systematize electronic documentation and to strengthen the quality of reporting.

### Pre-planned data collection

Our team has been using standardized instruments and electronic databases to systematically collect and store clinical information for more than a decade. These databases were developed not only to standardize and systematize the clinical documentation of the medical record, but also to foster analytics for quality assurance and to enable future research. This repository of information includes (under strict medical confidentiality) not only patient demographic data and existing medical comorbidities, but also a detailed "intake" assessment of the principal endometriosis-related symptoms and results of the self-administered validated Brazilian Portuguese version of HQoLife questionnaire. Data was assessed for research purposes on September 30, 2023.

## Non-menstrual pelvic symptoms assessment

The primary outcomes included four major non-menstrual pelvic symptoms: DDyspareunia, NM Pelvic Pain, NM Dyschezia and NM Dysuria, and the participants were instructed to report their symptoms as representative of the prior six months [25]) on a self-reported 11-point (0–10) numeric rating scale (NRS), in which pain can be hierarchically categorized as none/mild (0–3), moderate/tolerable (4–6), or severe (7–10) [32].

The questions for DDyspareunia asked: "Have you had pain during sexual intercourse in the last six months? If yes, is this pain at the beginning of penetration or during deep penetration?". The participants could check "not applicable", if they have not had sexual intercourse in the prior six months. NM Pelvic Pain was assessed if the patient had recurrent or constant pain in the lower abdomen, which is considered chronic if it had occurred for at least six months [33]. NM Dyschezia was assessed if the patients had discomfort or pain during defecation, outside of the menstrual period. NM Dysuria was assessed if the patient had a complaint of burning or other discomfort during micturition outside of the menstrual period.

## Health-related quality of life questionnaires

Different tools have been used to assess and quantify HQoLife in women with endometriosis. The Short Form 36 (SF36) and Endometriosis Health Profile 30 (EHP30) questionnaires have been, respectively, the most used and most specific instruments to assess HQoLife in women with endometriosis [34,35]. For clinical practice, these two questionnaires perform better overall with regard to their strengths and weaknesses when compared to other scales [36]. Both questionnaires were originally standardized on a scale from 0 to 100. However, a 100 score on the SF36 scale indicates the best health status, while a 100 score on the EHP30 scale indicates the worst health status, that is, they are inverse scales. In this study, both SF36 and EHP30 questionnaires were applied, but the SF36 scale was purposely inverted so that the value 100 represents the worst status in both questionnaires.

**The Short Form 36.** The SF36 is a multipurpose short-form health survey with 36 questions. It is a generic measure, as opposed to one that targets a specific age, disease, or treatment group. The SF36 is a self-reported questionnaire that yields an eight-scale profile of scores, encompassing eight domains/concepts of health, and hypothesized to form two distinct higher-ordered clusters according to the physical and mental health variance that they have in common: the Physical Component Summary and the Mental Component Summary [37] (Table 1). The SF36 is a valid measure for evaluating endometriosis HQoLife [38]. In this study, we used the Brazilian Portuguese version of the SF36 validated by Ciconelli et al, in 1999 [39].

**The Endometriosis Health Profile 30.** The EHP30 questionnaire is a self-administered instrument to measure the HQoLife of women with endometriosis [33]. We used the Brazilian Portuguese version of EHP30 validated by Mengarda et al, in 2008 [40]. Although both the SF36 and EHP30 have specific domains for pain, only the EHP30 questionnaire includes specific questions about dyspareunia and infertility (Table 2).

## Statistics

Bivariate Pearson's correlation analysis was used to express the strength of association between two scale variables. Every domain of both SF36 and EHP30 questionnaires was separately evaluated as a dependent variable (outcome) in an exploratory multiple linear regression model. In this study, the statistical models were not used with predictive purposes, but just to compare the relative contribution of explanatory variables (correlates). No explanatory variable was removed (i.e., no backward selection methods) because the objective of the study was not models optimization, but rather to assess the symptoms altogether in a multivariate context. Charts and statistics were developed using IBM® SPSS® Statistics Version 29.0.0.0–241 (IBM Corp., Armonk, NY, USA). Alpha threshold was set to 0.05. There was no treatment for missing values; the analyzes only included cases with actual verified values. Variance Inflation Factor (VIF) values ≥ 2.5 was conservatively considered indicative of collinearity [41].

**Table 1. Simplified structure of the Short Form 36 (SF36) questionnaire.**

| | | |
|---|---|---|
| Physical Component Summary | Physical Functioning SF36PF | 10 questions about how the patient's health limits her (1-A Lot, 2-A Little or 3-Not at All) in making 10 decreasing levels of activities, from (1) Vigorous ones: such as running, lifting heavy objects, participating in strenuous sports, (2) Moderate ones: such as moving a table, pushing a vacuum cleaner, bowling, or playing golf, (3) Lifting or carrying groceries, (4) Climbing several flights of stairs, (5) Climbing one flight of stairs, (6) Bending, kneeling, or stooping, (7) Walking more than a mile, (8) Walking several blocks, (9) Walking one block and (10) Bathing or dressing by herself. |
| | Physical Role Functioning SF36PRF | 4 questions regarding the past 4 weeks, about if the patient's physical health has been caused (yes/no) problems with work or other regular daily activities, including (1) the reduction of the time spent on them, (2) less accomplishment than desired, (3) limitation in any of them and (4) difficulty in performing them. |
| | Bodily Pain SF36 BP | 2 questions about (1) pain intensity (from 1-None to 6-Very Severe) and (2) its interference with work (whether outside the home or housework) from 1-Not at all to 5-Extremely, that the patient has been feeling during the past 4 weeks. |
| | General Health Perceptions SF36GHP (§) | 5 questions about (1) how the patient would say her health is, from 1-excellent to 5-poor. The others 4 questions are composed by statements and its levels of patient's agreement, from 1-"Definitely True" to 5- "Definitely False": (1) If the patient thinks she gets sick easier than the others, (2) if she thinks she is as healthy as anybody else, (3) if she expects her health gets worse and (4) if she believes her health is excellent. |
| | Vitality SF36VIT (§) | 4 questions, regarding the past 4 weeks, about the frequency which (1) did the patient feel full of pep, (2) if she had a lot of energy, (3) if she felt worn out and (4) if she felt tired. Answers graded into 6 levels, from 1-All of the Time to 6-None of the Time. |
| Mental Component Summary | Social Role Functioning SF36SRF (#) | 2 questions, regarding the past 4 weeks, (1) to what extent, from 1-Not at all to 5-Extremely and; (2) how much time spent, from 1-All of the time to 5-None of the time, have the patient's physical health or emotional problems interfered in normal social activities with family, friends, neighbors, or groups. |
| | Emotional Role Functioning SF36ERF | 3 questions, regarding the past 4 weeks, if the patient had (yes/no) problems with work or other daily activities as a result of emotional problems (such as feeling depressed or anxious): (1) If she cut down the amount of time spent in these activities, (2) if she accomplished less than she would like and (3) if she didn't do these activities as carefully as usual. |
| | Mental Health SF36MH | 5 questions (from 1-All of the Time to 6-None of the Time), about how much of the time, during the past 4 weeks, (1) the patient has been nervous, (2) if she has felt so down in the dumps that nothing could cheer her up, (3) if she has felt calm and peaceful, (4) if she has felt downhearted and blue and (5) if she has been a happy person. |

There is one question that doesn't take part in the calculation of any domain, being used only to assess how much better or worse the individual's general health is compared to one year ago, from 1- Much better to 5- Much worse. The SF36PF, SF36PRF and SF36 BP scales correlate most highly with the physical component and contribute most to the scoring of the Physical Component Summary measure. The mental component correlates most highly with the SF36MH, SF36ERF and SF36SRF scales, which also contribute most to the scoring of the Mental Component Summary measure. Three of the scales have noteworthy correlations with both components: the SF36VIT scale correlates substantially with both; SF36GHP correlates with both, but higher with the Physical Component Summary; and SF36SRF correlates much higher with Mental Component Summary [37].

§Also significantly correlated with the components of the Mental Component Summary.

#Also significantly correlated with the components of the Physical Component Summary.

## Ethics statement

The research protocol was approved by an institutional review board (Research Ethics Committee of the Oswaldo Cruz Institute Foundation-CAAE 07885019.8.0000.5269 IFF-FIOCRUZ) in accordance with the 1964 Declaration of Helsinki and subsequent reviews. Patients who might have declined to take part in the study would have received the same care as the patients who gave their consent to take part in the study.

The proposal to use (retrospectively) data from participants who had been evaluated from 1/January/2018 was approved by the Research Ethics Committee (CEP-IFF-Fiocruz) on 21/February/2019. As this is a retrospective study, the need for specific written consent for participants attended before 21/February2019 was waived by the Research Ethics Committee (CEP-IFF-Fiocruz). However, written informed consent was also obtained from them. In resume, written informed consent was obtained from all participants included in this study.

**Table 2. Simplified structure of the Endometriosis Health Profile 30 (EHP30) questionnaire.**

| Core | Pain EHP30Pain | 11 questions: 1- "Felt unable to cope with the pain?"; 2- "Been unable to do jobs around the home because of the pain?"; 3- "Found it difficult to stand because of the pain?"; 4-. "Found it difficult to sit because of the pain?"; 5- "Found it difficult to walk because of the pain?"; 6- "Been unable to sleep properly because of the pain?"; 7- "Found it difficult to exercise or do the leisure activities you would like to do because of the pain?"; 8- "Had to go to bed/lie down because of the pain?"; 9- "Lost appetite/unable to eat because of the pain?"; 10- "Been unable to go to social events because of the pain?"; 11- "Been unable to do the things you want to because of the pain?". |
|---|---|---|
| | Control and Powerlessness EHP30CPower | 6 questions: 1- "Generally felt unwell?"; 2- "Felt frustrated as symptoms are not getting better?"; 3- "Felt frustrated as not able to control symptoms?"; 4- "Felt unable to forget symptoms?"; 5- "Felt symptoms rule your life?"; 6- "Felt symptoms are taking away your life?". |
| | Social Support EHP30SSupp | 4 questions: 1- "Felt unable to tell people how you feel?"; 2- "Felt alone?"; 3- "Felt others do not understand what you are going through?"; 4- "Felt as though others think you are moaning?". |
| | Emotional Well-being EHP30EWbeing | 6 questions: 1- "Felt depressed?"; 2- "Felt weepy/tearful?"; 3- "Felt miserable?"; 4- "Had mood swings?"; 5- "Felt bad or short tempered?"; 6- "Felt violent or aggressive?". |
| | Self-image EHP30SImage | 3 questions: 1- "Felt frustrated as you cannot always wear the clothes you would choose?"; 2- "Felt your appearance has been affected?"; 3- "Lacked confidence?". |
| Modular | Sexual Intercourse EHP30Sex | 5 questions: 1- "Experienced pain during or after intercourse?"; 2- "Felt worried about having intercourse because of the pain?"; 3- "Avoided intercourse because of the pain?"; 4- "Felt guilty about not wanting to have intercourse?"; 5- "Felt frustrated because you cannot enjoy intercourse?". |
| | Work EHP30Work | 5 questions: 1- "Had to take time off work because of the pain?"; 2- "Been unable to carry out duties at work because of the pain?"; 3- "Felt embarrassment about symptoms at work?"; 4- "Felt guilty about taking time off work?"; 5- "Felt worried about not being able to do your job?". |
| | Feelings about the Medical Profession EHP30FMed | 4 questions: 1- "Felt the doctor(s) seem not doing anything for you?"; 2- "Felt doctor(s) think it is all in your mind?"; 3- "Felt frustrated at the doctor(s) lack of knowledge about endometriosis?"; 4- "Felt like you are wasting the doctor's time?". |
| | Infertility EHP30Infert | 4 questions: 1- "Felt worried about the possibility of not having children/more children?"; 2- "Felt inadequate because you may not/ have not been able to have children/ more children?"; 3- "Felt depressed at the possibility of not having children/ more children?"; 4- "Felt as though the possibility/inability to conceive has put a strain on your personal relationships?" |
| | Relationship with Children EHP30RChild | 2 questions: 1- "Found it difficult to look after your child/children?"; 2- "Been unable to play with your child/ children?". |
| | Feelings about Treatment EHP30FTreat | 3 questions: 1- "Felt frustrated because treatment is not working?"; 2- "Found it difficult coping with side effects of treatment?"; 3- "Felt annoyed at the amount of treatment you have had to have?". |

The Core questionnaire includes 30 questions and five domains. Six modular parts consisting of 23 questions were developed to measure some health status areas that may not affect all patients with endometriosis, but may be particularly relevant in specific cases only [34]. All answers from both core and modular EHP30 questionnaires have five levels: Never, Rarely, Sometimes, Often and Always.

## Results

Overall, the sample was composed mainly of healthy nulliparous European descent Brazilian women with steady partners who are occasional drinkers, non-smokers, not obese, with a higher education level and middle-class income (Table 3). Of the 369 cases, 12 women did not answer the question about DDyspareunia, 1 woman did not answer the question about NM Dyschezia, and 2 women did not answer the question about NM Dysuria. The prevalence of moderate/severe pain scores (NRS>3) was 53.8% for DDyspareunia, 58.5% for NM Pelvic Pain, 22.8% for NM Dyschezia and 8.4% for NM Dysuria. A total of 167 women (45.9%) reported chronic pelvic pain (NM Pelvic Pain lasting for 6 months or more). There was a significant positive correlation between NRS scores and duration of NM Pelvic Pain (P<0.001). Despite the weak linear correlation among the pain symptoms, the sample size allowed us to identify a statistical significance for each relationship (Fig 1).

**Table 3. Characteristics of the sample (N = 369).**

| | | N | (%) |
|---|---|---|---|
| **Ethnicity** | Asiatic | 3 | 0.8 |
| **(self reported)** | African descent | 38 | 10.3 |
| | European descent | 262 | 71.0 |
| | Brazilian Indigenous | 1 | 0.3 |
| | Mixed | 65 | 17.6 |
| **Partner** | Never | 69 | 18.7 |
| **(stable relationship)** | Not currently | 35 | 9.5 |
| | Yes | 265 | 71.8 |
| **Schooling** | High school | 2 | 0.5 |
| | High school (completed) | 28 | 7.6 |
| | College | 38 | 10.3 |
| | College (completed) | 134 | 36.3 |
| | Post-grad | 17 | 4.6 |
| | Post-grad (completed) | 150 | 40.7 |
| **Alcohol intake** | Never | 233 | 63.1 |
| | Once or twice a week | 122 | 33.0 |
| | 3 times a week or more | 14 | 3.9 |
| **Obstetrics** | No pregnancy | 233 | 63.1 |
| | Live birth (1 or more) | 103 | 27.9 |
| | Infertility | 138 | 37.7 |
| | ART in the past (1 or more) | 28 | 7.6 |
| **Income (U$/year)** | < 10,000 | 5 | 1.4 |
| | 10 to 20,000 | 52 | 14.3 |
| | 20 to 50,000 | 131 | 36.0 |
| | 50 to 100,000 | 118 | 32.4 |
| | > 100,000 | 58 | 15.7 |
| | Not informed | 5 | |
| **Conditions** | HAS | 23 | 6.3 |
| | DM | 8 | 2.2 |
| | Hysterectomy in the past | 8 | 2.2 |
| **Physical activity** | Never | 173 | 47.0 |
| | Once or twice a week | 57 | 15.5 |
| | 3 times a week or more | 138 | 37.5 |
| | Not informed | 1 | |
| **Smoking** | Never | 344 | (94.0) |
| | 5 or + cigarettes a day | 8 | (2.2) |
| | < 5 cigarettes a day | 14 | (3.8) |
| | Not informed | 3 | |
| | | **25th** | **Median** | **75th** |
| **Menarche (age in years)** | 11 | 12 | 13 |
| **Height (cm)** | 1.59 | 163 | 1.70 |
| **BMI (Kg. m-2)** | 22.0 | 24.0 | 27.0 |
| **Age (years)** | 31 | 36 | 41 |

Alcohol consumption: days a week. Income landmarks represent the total annual household income and were based on approximated values in December 2022. Physical activity: days per week. ART (Assisted Reproductive Technology) includes all fertility treatments in which either eggs or embryos are handled; the most common was *in vitro* fertilization. Infertility: failure to achieve a successful pregnancy after 12 months or more of regular unprotected intercourse [42], or due to impaired reproductive capacity individually or with your partner [43].

 

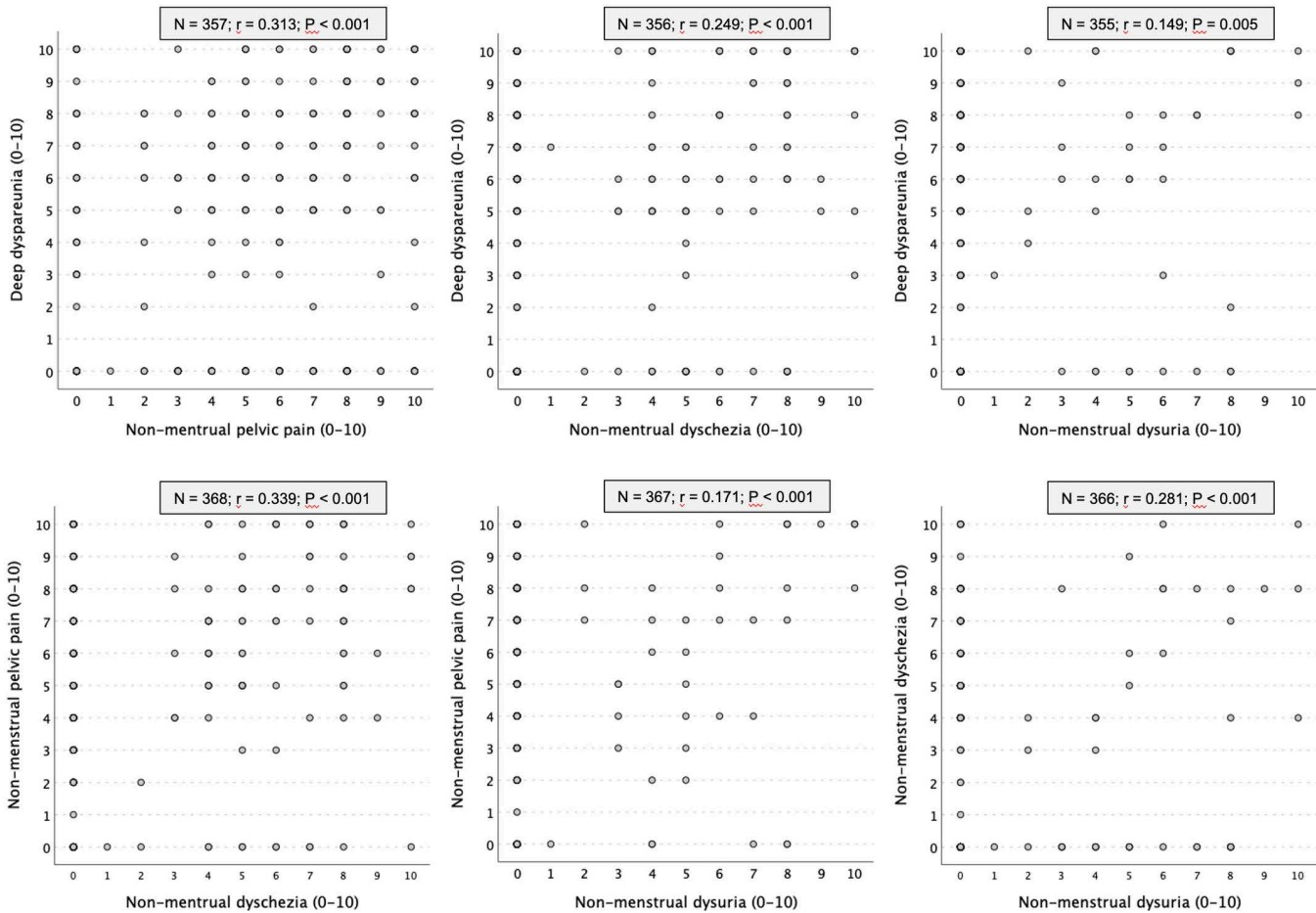

**Fig 1. Scatter plot and bivariate Pearson correlation analysis showing weak linear relationships among the pain symptoms.** A correlation coefficient between 0.25 and 0.50 was considered a "weak" correlation between two variables. The severity of pain was assessed on a self-reported 11-point (0 to 10) numeric rating scale. N: number of cases included in each analysis. r: Pearson Correlation Coefficient. P: statistical significance of "r". Fifteen of the 369 cases were excluded because they did not answer the questions about deep dyspareunia (12), about non-menstrual dyschezia (1), or about non-menstrual dysuria (2).

An overview of the HQoLife domains of the SF36 and EHP30 full questionnaires was shown in Fig 2. As the EHP30 questionnaire has usually been answered after the SF36 in the preoperative clinical evaluation routine at CRISPI, some patients completed only the first one (no specific reason detected). There was no significant correlation between women's age and any of the domains of the SF36 or EHP30 questionnaires. Thus, the variable "age" was not included in the regression models. [SF36 and EHP30 individual raw scores are found in S1 File.]

Purposely using DDyspareunia, NM Pelvic Pain, NM Dyschezia and NM Dysuria as the only explanatory variables, specific exploratory multiple linear regression models were tested for the 19 different HQoLife domains; results were exposed in Table 4. Statistically, the coefficient of determination ($R^2$) is a measure that represents the proportion of the variance for the outcome that is explained by the correlates; the higher the $R^2$, the better the model fits the data. Its value can only be between 0 and 1, where 0 indicates that the outcome cannot be explained by any of the explanatory variables and 1 indicates that the outcome can be explained without error by them. Most variables did not present a normal distribution (Shapiro-Wilk test). Although theoretically it would be possible to transform the variables to normality, the sample size

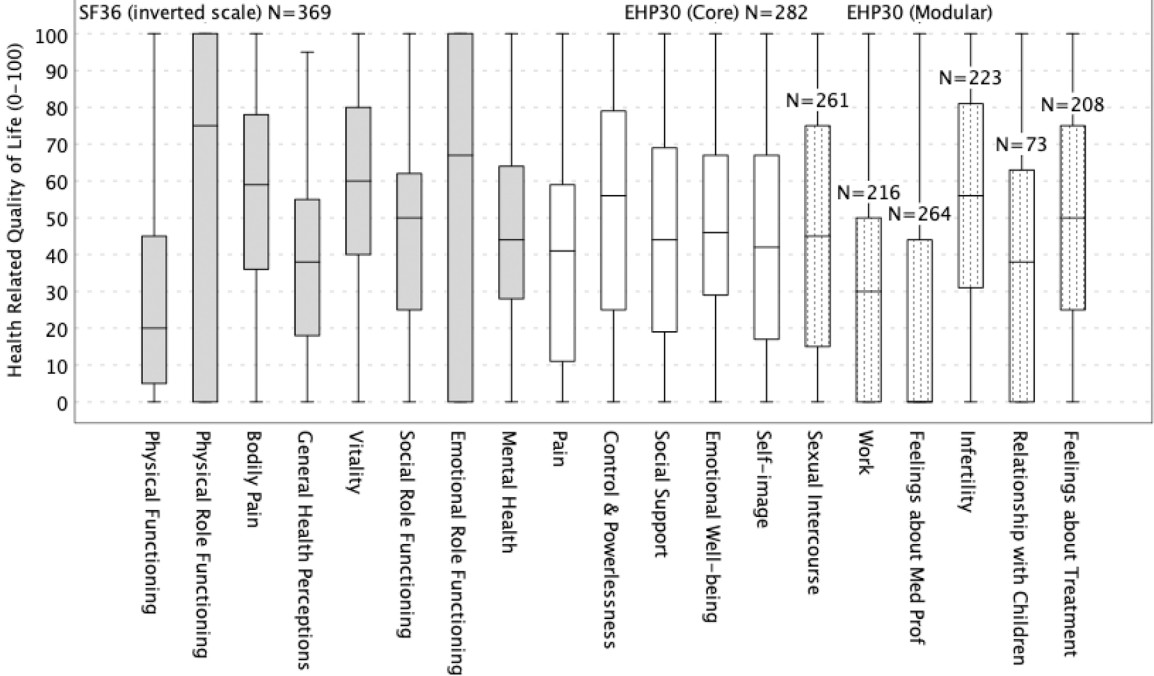

**Fig 2. Boxplot showing scores (quartiles) of the different health-related quality of life scales.** Both SF36 and EHP30 questionnaires were originally standardized on a scale of 0–100. However, paradoxically, a score of 100 on the SF36 scale indicates the best health status, while a score of 100 on the EHP30 indicates the worst health status. To make the comparison visually easier, the SF36 scale was purposely inverted so that the value 100 represents the worst status in both questionnaires. As the EHP30 questionnaire was answered after the SF36, 87 patients completed only the first one. The EHP30 optional supplementary module Relationship with Child/Children included only 73 women; this gap represents the prevalence of nulliparous status in the sample.

was considered sufficient for using natural raw data. Importantly, using transformed data makes the comparison of the regression coefficients (B) less natural for most readers - the main objective of this study. In all models tested, VIF values were between 1.06 and 1.21. [Details of the linear regression analyses are found in S2 File.]

In this study, although 17 of the 19 tested models were significant and most angular coefficients were positive and statistically different from zero, all models showed a low goodness-of-fit, which is in accord to R2 values taking into account these explanatory variables. The intercept (the constant of the equation) represents the mean value of the outcome when all correlates are equal to zero. In this study, the intercept was a positive value statistically different from zero in all models (P<0.001).

From the perspective of the SF36 questionnaire, NM Pelvic Pain and NM Dyschezia were significant correlates for all HQoLife domains whereas DDyspareunia was a significant correlate for 5 of the 8 SF36 domains (most from the Physical Component Summary). The values of the angular coefficients for DDyspareunia were comparatively smaller than those verified for the NM Pelvic Pain and NM Dyschezia. NM Dysuria was not a significant correlate for any of the SF36 domains.

From the perspective of the EHP30 questionnaire, NM Pelvic Pain and NM Dyschezia were significant correlates for all five domains of the core instrument whereas DDyspareunia was a significant correlate only for 2 of them. The angular coefficients for DDyspareunia were smaller than those observed for NM Pelvic Pain and NM Dyschezia in all models of the core instrument. When the models for each scale of EHP30's modular instrument were tested individually, NM Pelvic Pain was a significant correlate for 4 of the 6 domains, and it was the only significant correlate for the EHP30RwC and

Table 4. Multiple linear regression models using non-menstrual endometriosis-related pain symptoms (deep dyspareunia, non-menstrual pelvic pain, non-menstrual dyschezia and non-menstrual dysuria) as independent correlates for different domains of the health-related quality of life.

| | Model | Constant | Dyspareunia | NM Pelvic Pain | NM Dyschezia | NM Dysuria |
|---|---|---|---|---|---|---|
| | Adjusted R2 [P] | B (CI 95%) [P] | B (CI 95%) [P] | B (CI 95%) [P] | B (CI 95%) [P] | B (CI 95%) [P] |
| **SF36 [N=354]** | | | | | | |
| SF36PF | 0.187 [<0.001] | 12.9 (8.8/16.9) [<0.001] | 0.9 (0.3/1.6) [0.007] | 1.5 (0.8/2.2) [<0.001] | 2.2 (1.3/3.1) [<0.001] | 0.3 (-1.0/1.6) [0.658] |
| SF36PRF | 0.174 [<0.001] | 32.8 (25.7/40.0) [<0.001] | 1.0 (-0.2/2.1) [0.091] | 3.0 (1.8/4.3) [<0.001] | 3.0 (1.3/5.6) [<0.001] | 2.1 (-0.2/4.4) [0.069] |
| SF36 BP | 0.256 [<0.001] | 34.0 (30.0/38.2) [<0.001] | 1.1 (0.4/1.8) [0.001] | 2.3 (1.5/3.0) [<0.001] | 2.0 (1.1/2.9) [<0.001] | 0.7 (-0.6/2.0) [0.289] |
| SF36GHP | 0.176 [<0.001] | 26.4 (22.5/30.3) [<0.001] | 0.8 (0.2/1.4) [0.014] | 1.9 (1.2/2.6) [<0.001] | 1.3 (0.5/2.2) [0.003] | 0.0 (-1.2/1.3) [0.949] |
| SF36VIT | 0.166 [0.001] | 46.2 (42.1/50.3) [<0.001] | 0.8 (0.2/1.5) [0.016] | 1.7 (0.9/2.4) [<0.001] | 1.9 (1.0/2.9) [<0.001] | -0.4 (-1.7/0.9) [0.520] |
| SF36SRF | 0.177 [<0.001] | 27.9 (23.2/32.6) [<0.001] | 1.3 (0.5/2.1) [0.001] | 1.6 (0.7/2.4) [<0.001] | 2.5 (1.4/3.6) [<0.001] | -0.6 (-2.1/0.9) [0.423] |
| SF36ERF | 0.106 [<0.001] | 40.7 (33.2/48.1) [<0.001] | 0.3 (-0.9/1.5) [0.674] | 2.6 (1.3/4.0) [<0.001] | 3.3 (1.6/5.0) [<0.001] | -1.5 (-3.9/0.9) [0.221] |
| SF36MH | 0.158 [<0.001] | 34.8 (31.1/38.5) [<0.001] | 0.5 (-0.1/1.1) [0.119] | 1.7 (1.1/2.4) [<0.001] | 1.5 (0.7/2.3) [<0.001] | -0.6 (-1.8/0.6) [0.321] |
| **EHP30 (core) [N=271]** | | | | | | |
| EHP30Pain | 0.244 [<0.001] | 21.0 (16.1/26.0) [<0.001] | 0.8 (-0.2/1.6) [0.057] | 2.6 (1.7/3.4) [<0.001] | 2.4 (1.3/3.5) [<0.001] | -0.5 (-2.1/1.1) [0.554] |
| EHP30CPower | 0.250 [<0.001] | 30.4 (24.7/36.0) [<0.001] | 1.0 (0.0/1.9) [0.041] | 3.2 (2.2/4.2) [<0.001] | 2.5 (1.2/3.8) [<0.001] | -0.8 (-2.7/1.1) [0.397] |
| EHP30SSupp | 0.190 [<0.001] | 27.7 (22.1/33.2) [<0.001] | 0.9 (0.0/1.8) [0.045] | 2.6 (1.6/3.6) [<0.001] | 2.0 (0.7/3.3) [0.002] | -1.1 (-2.9/0.8) [0.255] |
| EHP30EWbeing | 0.179 [<0.001] | 34.2 (29.4/38.9) [<0.001] | 0.5 (-0.3/1.3) [0.220] | 2.2 (1.3/3.1) [<0.001] | 2.0 (0.9/3.1) [0.001] | -1.5 (-3.0/0.1) [0.066] |
| EHP30SImage | 0.140 [<0.001] | 27.4 (21.3/33.4) [<0.001] | 0.5 (-0.5/1.5) [0.325] | 2.5 (1.4/3.6) [<0.001] | 2.0 (0.6/3.4) [0.006] | -2.8 (-4.8/-0.8) [0.005] |
| **EHP30 (modular)** | | | | | | |
| EHP30Sex [N=253] | 0.414 [<0.001] | 18.1 (12.7/23.6) [<0.001] | 4.9 (4.0/5.8) [<0.001] | 1.3 (0.4/2.3) [0.006] | 0.4 (-0.8/1.7) [0.692] | 0.6 (-1.1/2.3) [0.656] |
| EHP30Work [N=209] | 0.190 [<0.001] | 14.3 (8.3/20.3) [<0.001] | 1.2 (0.2/2.2) [0.018] | 2.3 (1.2/3.4) [<0.001] | 1.8 (0.4/3.3) [0.013] | -0.1 (-2.3/2.0) [0.916] |
| EHP30FMed [N=253] | 0.016 [0.096] | 17.2 (10.7/23.7) [<0.001] | 0.5 (-0.6/1.6) [0.381] | 0.5 (-0.7/1.7) [0.447] | 1.4 (-0.1/2.9) [0.058] | -1.1 (-3.2/1.0) [0.295] |
| EHP30Infert [N=215] | 0.008 [0.220] | 49.4 (42.3/56.5) [<0.001] | 0.8 (-0.4/1.9) [0.205] | 0.2 (-1.1/1.5) [0.742] | 0.7 (-1.0/2.4) [0.410] | -2.1 (-4.4/0.2) [0.076] |
| EHP30RChild [N=71] | 0.088 [0.038] | 22.2 (9.9/34.4) [<0.001] | -0.6 (-2.8/1.6) [0.603] | 2.9 (0.6/5.2) [0.016] | 1.5 (-1.6/4.5) [0.346] | 1.0 (-4.3/6.3) [0.703] |
| EHP30FTreat [N=198] | 0.094 [<0.001] | 35.4 (28.5/42.3) [<0.001] | -0.3 (-1.4/0.9) [0.655] | 2.6 (1.4/3.9) [<0.001] | 1.0 (-0.6/2.6) [0.210] | -0.9 (-3.2/1.5) [0.469] |

Short-form 36 (SF36) domains: Physical Functioning (SF36PF), Physical Role Functioning (SF36PRF), Bodily Pain (SF36 BP), General Health Perceptions (SF36GHP), Vitality (SF36VIT), Social Role Functioning (SF36SRF), Emotional Role Functioning (SF36ERF) and Mental Health (SF36MH). Endometriosis Health Profile 30 (EHP30) Core questionnaire: Pain (EHP30Pain), Control and Powerlessness (EHP30CPower), Social Support (EHP30SSupp), Emotional Well-being (EHP30EWbeing) and Self-image (EHP30SImage). EHP30 modular questionnaire: Sexual Intercourse (EHP30Sex), Work (EHP30Work), Feelings about the Medical Profession (EHP30FMed), Infertility (EHP30Infert), Relationship with Children (EHP30RChild), Feelings about Treatment (EHP30FTreat). The SF36 scale was purposely inverted so that the value 100 represents the worst status in both questionnaires. Intercept: constant. R2 (adjusted coefficient of determination): represents the proportion of the variance for the dependent variable that is explained by the explanatory variables. B (unstandardized coefficient): indicates how much the dependent variable varies with an independent variable when all other independent variables are held constant (it means that for each one unit increase in the referred correlate, there is a decrease in B units in the outcome). Cells with statistically significant results (P<0.05) are shaded. Variance Inflation Factor < 2.5 was used as a conservative threshold to consider that there is no collinearity in the regression analysis [41]; all models met these requirements with a good safety margin (VIF values were between 1.06 and 1.21).

EHP30FaT scales. DDyspareunia was a significant correlate for only two of the six domains, and not surprisingly it was the most important correlate for the EHP30Sex. NM Dyschezia was a significant correlate only for EHP30Work, which was also significantly affected by DDyspareunia and NM Pelvic Pain. With a negative value, NM Dysuria was the sole significant correlate for the EHP30SI.

## Discussion

### Overview

This study analyzed HQoLife and four distinct and independent non-menstrual pelvic symptoms in consecutive patients with endometriosis referred to a specialized service for candidacy for minimally invasive surgery. Multivariate exploratory analyses indicated that NM Pelvic Pain and NM Dyschezia may be the non-menstrual pelvic symptoms that most impact the HQoLife of women with endometriosis, an association that was observed in practically all domains of both the SF36 and the EHP30 questionnaires.

Despite the biological plausibility and statistical significance found in virtually all models, the significant correlates are far from being considered the main determinants of poor HQoLife scores. Indeed, the low explanatory power of the models – evidenced by the weak R2 values – suggests the existence of additional major correlates (i.e., factors not contemplated in the models). Moreover, the presence of a significant positive intercept in all models implies that some HQoLife impairment is expected for any hypothetical woman with endometriosis, even if all four correlates' scores are equal to zero.

A possible overestimation of the (weak) variance in HQoLife outcomes attributed to the four non-menstrual pelvic symptoms, however, should not be construed as an error in the interpretation of statistics. Rather, it reflects an intuition that non-menstrual pain does matter, albeit not very consistently in any single case. Thus, one should not be dismissive of modest values for the percentage of the variance in a dependent variable that the Independent (explanatory) variables explain collectively, provided there is statistical assurance that these values are significantly above zero and that the cumulative potential may be substantial [44].

**The burden of deep dyspareunia.** The variable DDyspareunia was a statistically significant independent correlate for most SF36 domains, but not the strongest one in any model. Also, its burden on the two main domains of the Mental Component Summary (SF36ERF and SF36MH) was not statistically significant. From the point of view of the EHP30, the burden of DDyspareunia on HQoLife also proved to be far less important than NM Pelvic Pain or NM Dyschezia, the strongest independent correlates. Unsurprisingly, the strongest effect of DDyspareunia was on Sexual Intercourse (EHP30Sex), as it measures specific HQoLife consequences of pain during sex.

Other factors such as anxiety, depression, poor sleep quality, and pelvic pain may also affect sexual dysfunction and HQoLife in women with endometriosis [45]. As a complete assessment of sexual function is beyond the scope of this study, it is important to highlight that a simple biometric approach for measuring DDyspareunia (NRS scores) is insufficient and suggest that a more elaborate strategy is needed for studies that focus on women's sexual life [46].

**The potential burden of non-menstrual pelvic pain.** In line with the findings of other authors [20,47], NM Pelvic Pain was the most prevalent non-menstrual pelvic symptom in this sample, and may have a significant impact on almost all HQoLife domains. Despite the existence of covariates, these findings support previous studies pointing to chronic pelvic pain as the most important endometriosis-related pain symptom affecting women's HQoLife [5,20,47].

**The potential burden of non-menstrual dyschezia.** Despite being less prevalent than NM Pelvic Pain and DDyspareunia, NM Dyschezia also was a major correlate for HQoLife, showing strength equivalent to NM Pelvic Pain in virtually all models tested. Actually, although endometriotic bowel lesions can alter bowel function by different mechanisms [48], women with endometriosis have higher rates of dyschezia [49] and irritable bowel syndrome [50] than unaffected women, regardless of the presence of bowel endometriotic lesions. This uncertainty can make it difficult, for example, to

discuss surgical treatment of bowel endometriosis based on expected results. Our findings support the hypothesis that the importance of endometriosis-related bowel symptoms has been underestimated in endometriosis assessment.

**The potential burden of non-menstrual dysuria.** NM Dysuria was the least prevalent of the NM pelvic symptoms and 89.4% of the sample reported NRS=0. Unexpectedly, NM Dysuria was a significant negative correlate for EHP30 Self-Image domain. As a negative coefficient implies a beneficial effect of NM Dysuria on a HQoLife domain and this association lacks biological plausibility, we considered that this finding occurred by chance.

### Issues beyond acyclic pelvic pain

Although pain reduction has been one of the most important goals to improve the HQoLife in women with endometriosis [51], the burden of other potential covariates should not be neglected. Given the low explanatory power of the multivariate models summarized in Table 4, we can hypothesize that at least three types of factors may account for most of the variance in HQoLife scores: (1) Non-modifiable factors, such as genetic predisposition or a personal history of environmental exposure to endocrine disruptors [52,53,54]; (2) Modifiable factors, which may include other painful or specific functional conditions that can be treated with specific medications, surgery or both [24]; and (3) Potential confounders, which encompass Nonspecific Bodily Painful Conditions (e.g.,. migraine, irritable bowel and painful bladder syndromes), auto-immune diseases, fibromyalgia [6,55–58], as well as Lifestyle Habits (e.g., diet [19], physical activity [59], family, domestic responsibilities, and social functioning [14,60], professional life [61,62,63]) and, maybe the most complex, Concomitant Psychopathological Comorbidities (e.g., depression, anxiety, and emotional distress), which show higher rates in women with endometriosis than in the general population [17,57,64–66].

Very recently, nociplastic pain has entered the medical vernacular to describe pain conditions that are neither neuropathic (caused by nerve damage) nor nociceptive (caused by ongoing inflammation and damage of tissues), but commonly experienced by people worldwide [67]. Nociplastic pain may be a consequence of central sensitization - a highly prevalent condition that also includes chronic pain, allodynia, hypersensitivity, hyperalgesia, mood disturbances and depression [68]. Although the mechanisms that underlie this type of pain are not entirely understood [67], identifying patients whose pain is complicated by central nervous system sensitization may be the key problem when conventional treatment does not completely alleviate the pain [69]. Central sensitization may be defined operationally as amplification of neural signaling within the central nervous system that elicits pain hypersensitivity [70]; this condition has often been overlooked in patients with endometriosis, as well as the need for a multidisciplinary, multimodal approach to endometriosis-related pain [71]. The presence of non-menstrual symptoms does suggest some degree of central sensitization, which may negatively impact HQoLife. In this study, the low values of the explanatory coefficients (R2) and the presence of a significant negative constant in the multivariate linear regression models reinforce the concept that many covariates exist, so that chronic pelvic pain deserves interdisciplinary diagnostic and therapeutic approaches.

### Limitations

**Study design.** The main limitation of this study may be inherent to its design. As an uncontrolled observational study, it allows the presence of covariates that make interpretation of the findings and accurate conclusions difficult, which is less likely to occur with longitudinal and controlled studies. For example, the potential influence of hormonal blockade (a common therapeutic strategy in patients with endometriosis), concomitant relevant conditions and other pain symptoms were not controlled for.

**External validity.** Regarding external validity, the possibility of selection (or referral) bias associated with access to care should be considered. The sample lacks representation of some important groups [72]: adolescents, postmenopausal women, women of low socioeconomic status, women who do not identify as European descent, and non-heterosexual women.

**Statistical treatment of the scales.** Another limitation is that there is no consensus in the statistical treatment of the scales: whether proportional or ordinal; linear or nonlinear [5,73]. Thus, since the pain and HQoLife scales used in this

study measure the subjects' perception of an intangible quantity, ideally they should not be treated as proportion variables due to their inherent non-linearity [74]. Therefore, the exploratory parametric statistics used in this study (Pearson bivariate correlations and linear regression models) must be interpreted cautiously and only to compare the relative burden of explanatory variables, and not for the purpose of making predictions.

**Assumptions of regression analysis.** In this study, a multiple linear regression models (a statistical model that uses a linear equation) was used to explore the relationships between a quantitative dependent variable (HQoLife domain) and 4 quantitative explanatory variables (4 different NM pain symptoms). Regression analysis have several assumptions about the relationship between variables; they include linearity, homoscedasticity and normality. Besides, collinearity (sometimes termed multicollinearity) is usually defined as when two or more independent variables included in the model are highly correlated so that the values of one can be accurately predicted by that of another, which could be a problem. Normality, linearity, and homoscedasticity of variables enhance the analysis even when the statistics are used purely descriptively, as in this study. However, although variables did not present a normal distribution, the Central Limit Theorem reassures us that, with sufficiently large sample sizes, sampling distributions of means are normally distributed regardless of the distributions of variables [75]. In the same way, collinearity was not a problem because all calculated VIF values were between 1.06 and 1.21 – considered problematic when VIF ≥ 2.5 [41].

Concerning the cases-to-correlates ratio, required sample size depends on some issues, including the desired power, alpha and beta error levels, number of correlates, and expected effect sizes; conservatively considering a medium-size relationship between the correlates and the dependent variable, alpha = 0.05 and beta = 0.20, the minimum calculated sample size for a model assessing 4 explanatory variables should be 82 individuals [75]. Therefore, the sample size can be considered large enough for what it proposes, that is, to compare the relative burden of 4 different non-menstrual pelvic symptoms on HQoLife.

Although it is biologically plausible that any non-menstrual pain causes impairments to HQoLife, no causal relationship should be proposed in this cross-sectional study. As previously highlighted, the statistical models aimed only to compare the covariates.

## What is this study for? Interpretation/ Generalizability

In terms of strengths, this study may be considered stronger than most retrospective studies because it relied on systematic data-gathering processes initiated over 10 years ago and expanded and refined since then, carried out consistently by clinician-researchers and institute leadership who were committed to outcomes research from the outset.

A better understanding of the potential impact of different types of non-menstrual pelvic pain on HQoLife can help clinicians and patients choose one treatment strategy versus another (including assisted reproductive technology versus surgery in endometriosis-related infertility) because the planning must be individualized according to patient symptoms [29]. For example, if hysterectomy and bilateral salpingo-oophorectomy appear to provide greater benefit than excisional endometriosis surgery alone for non-menstrual pelvic symptoms as well as for HQoLife [76], a woman with non-menstrual pelvic symptoms could consider the first option as it affords greater certainty that benefits will occur. Women with endometriosis who had surgery and reported high levels of depression, anxiety or stress at the time of hospital admission, can benefit from cognitive behavioral therapy [77]. Moreover, these patients may have large and significant improvement in HQoLife with psychological intervention, even in the setting of persistent severe NM Pelvic Pain [78]. In fact, despite knowledge gaps and complex possible cause-and-effect relationships, mental disorders and endometriosis have gone hand in hand in several studies [79,80,81,82]. Although psychological disorders can be a response to pain phenomena, they can also contribute to their increase [83]. Therefore, our findings support a broader individualized diagnostic investigation (including with complementary therapeutic approaches aimed at comorbid conditions such as nonspecific bodily pain, lifestyle habits and psychopathological conditions that negatively affect the HQoLife of women with endometriosis-related pain.

## Conclusion

In conclusion, while DDyspareunia was the most important non-menstrual pelvic symptoms for the EHP30Sex domain, NM Pelvic Pain and NM Dyschezia were the most important for overall HQoLife. Despite being statistically significant, the multivariate statistical models including only non-menstrual pelvic symptoms showed modest explanatory power. Thus, the analysis raised the likely contribution of a set of covariates, which also should be contemplated and addressed for a more holistic and exhaustive approach to promoting and maximizing HQoLife. Further research in clinical settings may help identify which of these covariates may best contribute to improving HQoLife of women with deep endometriosis.

## Supporting information

**S1 File. Quality of life scores (raw data).**
(XLSX)

**S2 File. Details of the linear regression analyses.**
(PDF)

## Acknowledgments

The authors thank Dr. Leigh J. Passman for reviewing this manuscript.

## Author contributions

**Conceptualization:** Marlon de Freitas Fonseca, Felipe Ventura Sessa, Claudio Peixoto Crispi Jr, Claudio Peixoto Crispi.

**Data curation:** Claudio Peixoto Crispi Jr, Claudio Peixoto Crispi.

**Formal analysis:** Marlon de Freitas Fonseca.

**Investigation:** Felipe Ventura Sessa, Bruna Rafaela Santos de Oliveira, Claudio Peixoto Crispi.

**Methodology:** Marlon de Freitas Fonseca.

**Software:** Marlon de Freitas Fonseca, Bruna Rafaela Santos de Oliveira.

**Supervision:** Marlon de Freitas Fonseca, Felipe Ventura Sessa, Claudio Peixoto Crispi Jr, Nilton de Nadai Filho, Rafael Ferreira Garcia.

**Validation:** Felipe Ventura Sessa, Claudio Peixoto Crispi Jr, Nilton de Nadai Filho, Rafael Ferreira Garcia.

**Writing – original draft:** Marlon de Freitas Fonseca.

**Writing – review & editing:** Felipe Ventura Sessa, Claudio Peixoto Crispi Jr, Nilton de Nadai Filho, Bruna Rafaela Santos de Oliveira, Rafael Ferreira Garcia.

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
