## [Decision Letter · Decision Letter 0]

21 Feb 2025

PONE-D-24-36216THE BURDEN OF DIFFERENT ENDOMETRIOSIS-RELATED NON-MENSTRUAL PELVIC SYMPTOMS ON WOMEN'S QUALITY OF LIFE: AN EXPLORATORY OBSERVATIONAL STUDYPLOS ONE

Dear Dr. Fonseca,

Thank you for submitting your manuscript to PLOS ONE. After careful consideration, we feel that it has merit but does not fully meet PLOS ONE’s publication criteria as it currently stands. Therefore, we invite you to submit a revised version of the manuscript that addresses the points raised during the review process.

We look forward to receiving your revised manuscript.

Kind regards,

Diego Raimondo

Academic Editor

PLOS ONE

Journal Requirements:

-https://pure.uvt.nl/ws/portalfiles/portal/266549/whoqolgroupbref1998.pdf

- https://doi.org/10.3390/ijtm2040041

In your revision ensure you cite all your sources (including your own works), and quote or rephrase any duplicated text outside the methods section. Further consideration is dependent on these concerns being addressed.

3. Please amend your manuscript to include your abstract after the title page.

5. Please include captions for your Supporting Information files at the end of your manuscript, and update any in-text citations to match accordingly. Please see our Supporting Information guidelines for more information: http://journals.plos.org/plosone/s/supporting-information .

Reviewers' comments:

Reviewer's Responses to Questions

**Comments to the Author**

1. Is the manuscript technically sound, and do the data support the conclusions?

Reviewer #1: Yes

Reviewer #2: No

Reviewer #3: Yes

2. Has the statistical analysis been performed appropriately and rigorously? 

Reviewer #1: Yes

Reviewer #2: No

Reviewer #3: Yes

3. Have the authors made all data underlying the findings in their manuscript fully available?

Reviewer #1: Yes

Reviewer #2: Yes

Reviewer #3: Yes

4. Is the manuscript presented in an intelligible fashion and written in standard English?

Reviewer #1: Yes

Reviewer #2: Yes

Reviewer #3: Yes

5. Review Comments to the Author

Reviewer #1: This manuscript presents the association between endometriosis-related pain duration and burden and poorer psychological well-being and health-related quality of life. Particular attention is paid to comparing the relative burden of different nonmenstrual pelvic symptoms (NMPSymptoms) on health-related quality of life (HRQOL).

Reviewer #2: Thank you for the opportunity to review your submission. Understanding the differential impacts of endometriosis-related symptomatology is an important research and clinical area. I have provided some specific feedback for your consideration below:

General:

1. The tile is quite long. It could be revised for reading ease.

2. The use of so many acronyms impacts on readability. I do not think you need to abbreviate so many of the terms.

Introduction:

1. Lines 9-12 - Please include citation for the statement.

2. Your introduction would benefit from a greater emphasis on your study rationale, aims, and hypotheses. These are lacking from the current submission. Your rationale needs more focus on the reasons and potential research/clinical gains for conducting this study.

Methods:

1. The SF36 table could be moved to supplementary data.

2. The EHP-30 table can also be moved to supplementary data file.

Results:

1. You have used cross-sectional data and so you need to use the term correlate not predictor.

2. Please report your data testing assumptions (e.g., VIF). Did you have to remove any variables because they violated statistical assumptions?

3. The reporting of the regression analyses requires further work. Please discuss the variance explained.

4. Please provide a rationale for excluding covariates? I think this creates problems for your results and discussion section.

Discussion:

1. The discussion is lacking in depth. There are also several causal statements made throughout and these will need to be reviewed as your data is cross-sectional in nature.

Reviewer #3: Dear Authors,

congratulations for your article.

Honestly, I believe it is very well written and I have just a few suggestions in order to give insight into some aspects that have not been adequately underlined.

Overall, the structure of the paper is correct and there is a great logicality, making it easily readable.

I only suggest to discuss the potential impact of NM symptoms into the management of patients potentially facing the need for surgery when thriving for pregnancy (http://dx.doi.org/10.3390/jcm13237349).

Eventually, which do you think is be the impact of CS in this kind of symptoms (http://dx.doi.org/10.1016/j.jmig.2022.10.007)?

6. PLOS authors have the option to publish the peer review history of their article (what does this mean? ). If published, this will include your full peer review and any attached files.

**Do you want your identity to be public for this peer review?** For information about this choice, including consent withdrawal, please see our Privacy Policy .

Reviewer #1: No

Reviewer #2: No

Reviewer #3: No

---

## [Author Response · Author response to Decision Letter 1]

10 Mar 2025

Journal Requirements:

Dear Reviewers,

Dear Editor,

We would like to thank you for this opportunity.

This study has undoubtedly been greatly improved thanks to the suggestions we have received.

As already mentioned, this manuscript was written for readers from different areas. Therefore, we have had a challenge: to present details (statistical rigor to obtain the results), while at the same time providing an easy and understandable reading of the discussion and conclusion.

Changes have been made to some abbreviations, some expressions and some mathematical details. A supplementary PDF file has been added. Some paragraphs have been rewritten to make them easier to read. A total of 10 references have been added to address reviewers' comments and suggestions.

We hope we have succeeded.

Thank you.

marlon & Col.

AUTHORES REPLY: Corrections were made according to PLOS ONE's style requirements. Thank you!

-https://pure.uvt.nl/ws/portalfiles/portal/266549/whoqolgroupbref1998.pdf

- https://doi.org/10.3390/ijtm2040041

In your revision ensure you cite all your sources (including your own works), and quote or rephrase any duplicated text outside the methods section. Further consideration is dependent on these concerns being addressed.

AUTHORES REPLY: It was corrected.

-https://pure.uvt.nl/ws/portalfiles/portal/266549/whoqolgroupbref1998.pdf

This publication was already in the reference list. It was correctly positioned in the manuscript. Please see 2nd paragraph of INTRODUCTION Section (below).

“Potentially influenced by biological, psychological and social factors, pain may be defined as an unpleasant sensory and emotional experience associated with, or resembling that associated with, actual or potential tissue damage [7]. Quality of life may be defined as individuals’ perceptions of their position in life in the context of the culture and value systems in which they live and in relation to their goals, expectations, standards and concerns; this definition reflects the view that quality of life refers to a subjective evaluation that is embedded in a cultural, social and environmental context [8]. Health-related quality of life (HQoLife) is generally considered a multidimensional assessment of how disease and treatment affect a patient’s sense of overall function and well-being [9].”

- https://doi.org/10.3390/ijtm2040041

This publication was not included in the reference list. The paragraph was rewritten so that there were no overlapping sentences (above).

3. Please amend your manuscript to include your abstract after the title page.

AUTHORES REPLY: It was corrected. Thank you!

AUTHORES REPLY: It was corrected. The topic "Ethics statement" (composed of 2 paragraphs) was properly placed in the end of METHODS section. Thank you!

AUTHORES REPLY: Two “In-Text Citations” were included in the manuscript (please see below) and the respective supporting information caption was included after de CONCLUSION Section.

(1) The sentence “The SF36 and EHP30 individual raw scores are found in S1 File” was included in the end of 2nd paragraph of Results section.

(2) The sentence “Details of the linear regression analyses are found in S2 File” was included in the end of 3rd paragraph of Results section.

Supporting information

S1 File. Quality of life scores (raw data).

(XLSX)

S2 File. Details of the linear regression analyses.

(PDF)

AUTHORES REPLY: The publications / citations were reviewed; they are correctly positioned in the manuscript. Following reviewers' suggestions, a total of 7 (seven) references were included.

Review Comments to the Author

Reviewer #1:

This manuscript presents the association between endometriosis-related pain duration and burden and poorer psychological well-being and health-related quality of life. Particular attention is paid to comparing the relative burden of different nonmenstrual pelvic symptoms (NMPSymptoms) on health-related quality of life (HRQOL).

AUTHORES REPLY: Thank you very much for your attention.

Reviewer #2:

Thank you for the opportunity to review your submission. Understanding the differential impacts of endometriosis-related symptomatology is an important research and clinical area. I have provided some specific feedback for your consideration below:

General:

1. The tile is quite long. It could be revised for reading ease.

AUTHORES REPLY: Thank you very much for your suggestion. The title has been shortened:

“Non-menstrual pelvic symptoms and women's quality of life: a cross-sectional observational study”.

2. The use of so many acronyms impacts on readability. I do not think you need to abbreviate so many of the terms.

AUTHORES REPLY: Thank you very much for your suggestion. All abbreviations were reviewed. Some were removed; others were modified to make them easier to read. To ensure maximum robustness in the results (which support the study), the abbreviations for the questionnaire scores were maintained in the Methods and Results sections (in parallel with the construction of the tables). However, the use of these details was avoided in the Discussion section.

Introduction:

1. Lines 9-12 - Please include citation for the statement.

AUTHORES REPLY: The Introduction section was rewritten (please see item 2, below); references were included.

2. Your introduction would benefit from a greater emphasis on your study rationale, aims, and hypotheses. These are lacking from the current submission. Your rationale needs more focus on the reasons and potential research/clinical gains for conducting this study.

AUTHORES REPLY: Thank you very much for your suggestion. This study was designed to bring together professionals / scholars from different areas. Thus, keeping the focus on this objective, the Introduction section was rewritten in 6 small paragraphs and 5 references were added in order to improve study rationale, hypotheses and aims.

In short:

Paragraph 1 introduces the condition "endometriosis".

Paragraph 2 introduces the 2 main concepts - pain and quality of life.

Paragraph 3 presents the idea that cyclical pain can evolve into chronic / acyclic pain.

Paragraph 4 highlights the existence of covariates, which can influence the outcome of a treatment (quality of life can be affected not only by painful processes, but also by emotional conditions).

Paragraph 5 highlights the importance of both healthcare professionals and patients understanding / knowing the potential risks and benefits of treatment.

Paragraph 6 highlights the importance of understanding about how different endometriosis pains are for different women and justifies the aim of the study (compare the relative burdens of different non-menstrual symptoms).

Methods:

1. The SF36 table could be moved to supplementary data.

AUTHORES REPLY: Please see below.

2. The EHP-30 table can also be moved to supplementary data file.

AUTHORES REPLY: Obviously, the SF36 and EHP30 tables can be presented as supplementary data. In other words, this will be done if the reviewers deem it best. However, before making this change, we would like to justify the reason for these tables to be in the body of the manuscript.

When we started studying quality of life in patients with endometriosis, we faced difficulties in understanding the domains of the questionnaires, since the studies usually do not present details of the questionnaire used (SF36, EHP30, etc.). Thus, readers cannot consult this information quickly if they are reading articles printed on paper. As stated in item 2 above (regarding Introduction section), this study was designed to bring together professionals / scholars from different areas. Actually, these 2 tables were created with a single objective: to allow (for example) a surgeon to quickly see how the SF36 and EHP30 variables were obtained (even if without a computer).

Results:

1. You have used cross-sectional data and so you need to use the term correlate not predictor.

AUTHORES REPLY: In the Methods section (first submitted version), we had written that "In this study, multiple linear regression models were used not exactly with predictive purposes, but just to compare the relative contribution of independent variables". However, we agree with you that the term "predictor" was not appropriate; we replaced it with "correlate" exactly as suggested. Thank you very much for your attention!

2. Please report your data testing assumptions (e.g., VIF). Did you have to remove any variables because they violated statistical assumptions?

AUTHORES REPLY: In view of the questions, we were forced to add more paragraphs in Methods, Results and Discussion sections. We hope they will be sufficient.

METHODS Section (topic Statistics):

“…In this study, the statistical models were not used with predictive purposes, but just to compare the relative contribution of explanatory variables (correlates). No explanatory variable was removed (i.e. no backward selection methods) because the objective of the study was not models optimization, but rather to assess the symptoms altogether in a multivariate context. […]. Variance Inflation Factor (VIF) values ≥ 2.5 was conservatively considered indicative of collinearity [41].”

Results section:

“Purposely using DDyspareunia, NM Pelvic Pain, NM Dyschezia and NM Dysuria as the only explanatory variables, specific exploratory multiple linear regression models were tested for the 19 different HQoLife domains; results were exposed in Table 4. Statistically, the coefficient of determination (R2) is a measure that represents the proportion of the variance for the outcome that is explained by the correlates; the higher the R2, the better the model fits the data. Its value can only be between 0 and 1, where 0 indicates that the outcome cannot be explained by any of the explanatory variables and 1 indicates that the outcome can be explained without error by them. Most variables did not present a normal distribution (Shapiro-Wilk test). Although theoretically it would be possible to transform the variables to normality, the sample size was considered sufficient for using natural raw data. Importantly, using transformed data makes the comparison of the regression coefficients (B) less natural for most readers - the main objective of this study. In all models tested, VIF values were between 1.06 and 1.21. [Details of the linear regression analyses are found in S2 File.]”

DISCUSSION Section / Limitations / Assumptions of regression analysis:

“In this study, a multiple linear regression models (a statistical model that uses a linear equation) was used to explore the relationships between a quantitative dependent variable (HQoLife domain) and 4 quantitative explanatory variables (4 different NM pain symptoms). Regression analysis have several assumptions about the relationship between variables; they include linearity, homoscedasticity and normality. Besides, collinearity (sometimes termed multicollinearity) is usually defined as when two or more independent variables included in the model are highly correlated so that the values of one can be accurately predicted by that of another, which could be a problem.

Normality, linearity, and homoscedasticity of variables enhance the analysis even when the statistics are used purely descriptively, as in this study. However, although variables did not present a normal distribution, the Central Limit Theorem reassures us that, with sufficiently large sample sizes, sampling distributions of means are normally distributed regardless of the distributions of variables [70]. In the same way, collinearity was not a problem because all calculated VIF values were between 1.06 and 1.21 – considered problematic when VIF ≥ 2.5 [41].

Concerning the cases-to-correlates ratio, required sample size depends on some issues, including the desired power, alpha and beta error levels, number of correlates, and expected effect sizes; conservatively considering a medium-size relationship between the correlates and the dependent variable, alpha = 0.05 and beta = 0.20, the minimum calculated sample size for a model assessing 4 explanatory variables should be 82 individuals [70]. Therefore, the sample size can be considered large enough for what it proposes, that is, to compare the relative burden of 4 different non-menstrual pelvic symptoms on HQoLife.

Although it is biologically plausible that any non-menstrual pain causes impairments to HQoLife, no causal relationship should be proposed in this cross-sectional study. As previously highlighted, the statistical models aimed only to compare the covariates.”

3. The reporting of the regression analyses requires further work. Please discuss the variance explained.

AUTHORES REPLY: The reporting of the regression analyses was now presented in 4 paragraphs - RESULTS Section (see below). Also, details about "the proportion of the variance for the dependent variable that is explained by the explanatory variables" are displayed in the caption to Table 1.

“ […] Purposely using DDyspareunia, NM Pelvic Pain, NM Dyschezia and NM Dysuria as the only explanatory variables, specific exploratory multiple linear regression models were tested for the 19 different HQoLife domains; results were exposed in Table 4. Statistically, the coefficient of determination (R2) is a measure that represents the proportion of the variance for the outcome that is explained by the correlates; the higher the R2, the better the model fits the data. Its value can only be between 0 and 1, where 0 indicates that the outcome cannot be explained by any of the explanatory variables and 1 indicates that the outcome can be explained without error by them. Most variables did not present a normal distribution (Shapiro-Wilk test). Although theoretically it would be possible to transform the variables to normality, the sample size was considered sufficient for using natural raw data. Importantly, using transformed data makes the comparison of the regression coefficients (B) less natural for most readers - the main objective of this study. In all models tested, VIF values were between 1.06 and 1.21. [Details of the linear regression analyses are found in S2 File.]

In this study, although 17 of the 19 tested models were significant and most angular coefficients were positive and statistically different from zero, all models showed a low goodness-of-fit, which is in accord to R2 values taking into account these explanatory variables. The inter

---

## [Editor Report · Decision Letter 1]

14 Mar 2025

Non-menstrual pelvic symptoms and women's quality of life: a cross-sectional observational study.

PONE-D-24-36216R1

Dear Dr. Fonseca,

We’re pleased to inform you that your manuscript has been judged scientifically suitable for publication and will be formally accepted for publication once it meets all outstanding technical requirements.

Kind regards,

Diego Raimondo

Academic Editor

PLOS ONE
---

## [Editor Report · Acceptance letter]

PONE-D-24-36216R1

PLOS ONE

Dear Dr. Fonseca,

I'm pleased to inform you that your manuscript has been deemed suitable for publication in PLOS ONE. Congratulations! Your manuscript is now being handed over to our production team.

Kind regards,

on behalf of

Dr. Diego Raimondo

Academic Editor

PLOS ONE